# Ecological Footprint Assessment of Concrete: Partial Replacement of Cement by Water Treatment Sludge and Stone Dust

Yakub Ansari [1], Dilawar Husain [2], Umesh Kumar Das [1], Jyotirmoy Haloi [3], Nasar Ahmad Khan [4], Ravi Prakash [5,*] and Mujahid Husain [6]

1   Department of Civil Engineering, University of Engineering and Management, Jaipur 303807, India
2   Department of Mechanical Engineering, Maulana Mukhtar Ahmad Nadvi Technical Campus, Mansoora Malegaon, Nashik 423203, India
3   Department of Civil Engineering, NIT, Silchar 788118, India
4   Department of Civil Engineering, Maulana Mukhtar Ahmad Nadvi Technical Campus, Mansoora Malegaon, Nashik 423203, India
5   Department of Mechanical Engineering, Motilal Nehru National Institute of Technology, Allahabad 211004, India
6   Department Civil Engineering, SSBT College of Engineering & Technology, Bambhori Jalgaon 425002, India
*   Correspondence: rprakash@mnnit.ac.in

**Abstract:** Currently, most concrete industries use conventional cement (Ordinary Portland Cement) as a binding material which involves natural resource depletion, colossal $CO_2$ emissions, and a huge energy supply. The present study addresses this critical issue by using stone dust (sun-dried and calcinated) and water treatment sludge (sun-dried and calcinated) to replace cement partly in M20-grade concrete production. The environmental impact of ready-mixed concrete (RMC) production with conventional cement and partially replaced cement by other cementitious material, i.e., stone dust and water treatment sludge in concrete, is assessed through ecological footprint (EF) indicator. Moreover, a novel sustainability index is proposed for ready-mixed concrete plants to scale the environmental impact of different types of concrete (or grades) on the sustainability scale (environmental, social, and economic sustainability). The results showed that the sun-dried water treatment sludge and sun-dried stone dust could effectively replace cement (15% by weight) in the concrete, with a comparable compressive strength over the M20 ready-mixed concrete. The EF of conventional M20 RMC is estimated to be 0.02295 gha/m³. The EF of concrete (with sun-dried water treatment sludge) is reduced by 13.14% of the conventional ready-mixed concrete. The Ecological Sustainability Index (ESI) of the ready-mixed concrete plant is estimated to be 718.42 \$/gha. Using water treatment sludge and stone dust in concrete production can be an innovative solution because it simultaneously solves the problem of waste disposal, large carbon emissions, cost, and high environmental impact.

**Keywords:** ready-mixed concrete; ecological footprint; stone dust; water treatment sludge

## 1. Introduction

Concrete has been one of the most prolific and widely used materials for building structures in modern industrialized society due to its robustness, strength, and low cost [1]. Moreover, it is also considered a versatile and durable material for climate-resilient construction [2]. Worldwide, around 32 billion tonnes of concrete are produced annually. Cement contributes around 10% of the concrete mass and at present it is produced at a rate of 4 billion tonnes per year globally [3]. The consumption of concrete has nearly tripled in the last 40 years, accounting for almost four tonnes per capita annually, and this demand is expected to rise further [3]. Although the benefits of concrete are plenty, it is considered

a significant threat to environmental sustainability due to its colossal carbon emissions, waste accumulation, and resource crisis [4,5].

Concrete, a synthetic rock, is prepared by adding sand and gravel to Ordinary Portland Cement (OPC), mixing them with water, and casting them into desired shapes or building blocks. OPC is the most critical part of concrete production due to carbon emissions. OPC manufacturing involves heating a mixture of limestone and clay at a temperature of up to 1500 °C in a kiln. This high-heating and simple reaction process, which is key to manufacturing cement, generates around 657 kg of carbon dioxide for producing one tonne of cement [1,3].

OPC contributes to roughly 7–10% of anthropogenic carbon dioxide ($CO_2$) emissions, consuming 2–3% of the global energy supply [6] and depleting 40–60% of natural material resources [7]. The cement industry is expected to grow at around 12–23% by 2050 as compared to the present production level [8,9], which will approximately increase global $CO_2$ emissions up to 11–15% [10] if cement production methods are not altered. It is already established that the rise in the earth's global temperature, rising sea levels, melting glaciers, and climatic changes are mainly attributed to anthropogenic greenhouse gas (GHG) emissions [3–8]. Additionally, $CO_2$ and other GHG emissions, dust, particulate matter, and mercury are also associated with cement and concrete production [3].

Due to these factors and growing global demand, concrete production is heavily regulated to reduce its negative environmental effects, mainly $CO_2$ emissions. The cement industry has been implementing various measures over the past few years to reduce the environmental impact of $CO_2$ emissions, including enhancing the efficiency of cement kilns, adopting renewable resources as substitutes for fossil fuels, and using supplementary cementitious materials to replace clinker [10]. For the construction industry, which utilizes over 50% of the earth's resources [11], one of the major challenges for attaining sustainable development is effective dematerialization, waste reduction, and recycling. This would also lessen the associated environmental impacts [11,12]. Thus, an alternative method for producing concrete with high sustainability index without compromising its physical properties becomes imminent. The most promising course to a large-scale reduction in carbon emissions comes from substituting cement components with a suitable alternative material, for example, supplementing cementitious materials with appropriate cementitious materials in order to replace a significant percentage of the cement [12].

The most appealing method is to utilize the inevitable waste generated in the industries as a fractional substitution of sand and cement, developing a low resource crisis and lower carbon footprint materials [1–3]. A recent study showed that stone dusts can be successfully used in concrete production up to 10% of cement replacement [13–15]. They concluded that the use of stone dust alleviates the environmental impact of concrete production and sustainable waste management options. Another, widely accepted cement replacement is the use of water treatment sludge (WTS) in concrete production. The WTS mainly comes from flocculation tanks, sedimentation tanks, and filter backwash wastewater [16]. It contains minerals, sand particles, a small amount of organic matter, and coagulants [17]. Although WTS is harmless in most cases compared to sewage sludge, WTS without harmless treatment is directly landfilled, which not only wastes land but also may lead to environmental problems such as secondary pollution [18]. WTS has binding properties and can partially replace cement without affecting its strength and workability. However, due to their limited reactivity, at substitution levels above 10–15%, it acts as a filler and not a cementitious material [16]. The environmental impact assessment of concrete with substitute cementitious materials must be assessed with available environmental techniques to accurately estimate their imprint on the planet. Ecological footprint indicator is one of the widely accepted and cumulative tool that can be used to assess the environmental impact of products, processes, and human activities, etc.

The ecological footprint study was created by Mathis and Rees for a quantitative assessment of processes and human activities impact on the environment [19]. The assessment tool can be used to evaluate the viability of various sustainable solutions for the

equitable distribution of the biocapacity of the region, nation, or planet. The ecological footprint indicator is distinctive: it incorporates all input resources. It converts them into a global hector (gha) parameter, is a measure of bio-productive land with world average productivity [20].

The present study focuses on assessing the ecological footprint (environmental impact) of M20 grade ready-mixed concrete as well as assess the cement replacement with stone dust and water treatment sludge. The cumulative footprint of energy, carbon, raw materials, and transportation is evaluated for normal concrete and ready-mix concrete at plant is evaluated for four different cases. The four cases are (i) calcinated WTS, (ii) sun-dried WTS, (iii) calcinated stone dust, and (iv) sun-dried stone dust. The ecological footprint of all the four cases with different percentage of backwash water sludge and stone dust were evaluated. The best case is said to be the one with minimum ecological footprint and optimum compressive strength. The main novelty of the present study includes two proposed methodologies: (1) An ecological footprint assessment methodology for ready mixed concrete, (2) sustainability indicator (novel Ecological Sustainability Index) for ready-mixed concrete plants.

The Ecological Sustainability Index (ESI) is a simple and effective tool that can represent the socio-economic benefit of industry (any type and scale) per unit of its biocapacity consumption. The remainder of the paper is organized as follows. Section 2 details the properties of materials used, ecological footprint analysis, and proposed ESI of conventional ready-mix concrete and partially cement replacement concrete. Section 3 discusses the description of ready-mix plant. Section 4 shows the discussion on results, and Section 5 summarizes the work of the present paper.

## 2. Materials and Methods

The ecological footprint analysis provides a chance to evaluate the global environmental effects of producing concrete. This research study is carried out to compute the ecological footprint of ready-mixed concrete production as well as partial replacement of cement with stone dust and water treatment backwash sludge in the ready-mixed concrete. The ecological footprint assessment of concrete production is as follows:

### 2.1. Ecological Footprint of Ready-Mixed Concrete ($EF_{RMC}$)

In this study, four parameters are considered to assess the ecological footprint of concrete production in concrete plant: (1) raw materials impact, (2) machinery/energy use and (3) labor, and (4) transportation. The flow chart of the ecological footprint of concrete production is depicted in Figure 1. The ecological footprint of ready-mixed concrete production has been estimated as Equation (1):

$$EF_{RMC} = EF_m + EF_e + EF_l + EF_t \tag{1}$$

where $EF_m$ represents the ecological footprint of raw materials use in concrete; $EF_e$ represents the ecological footprint of energy consumption in the concrete plant. $EF_l$ represents the ecological footprint of labor in concrete production and $EF_t$ represents the transportation of raw materials from their source to plant.

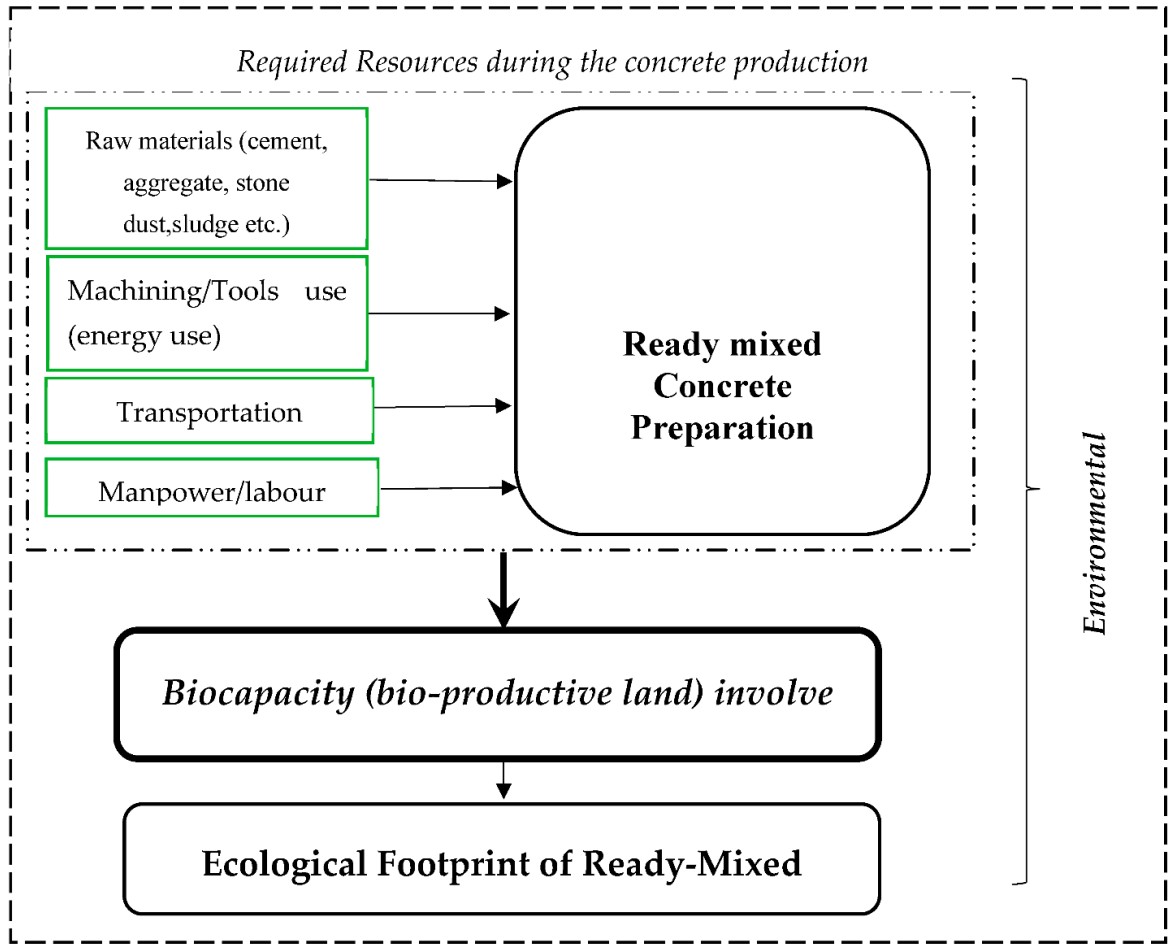

**Figure 1.** Flow diagram of ecological footprint assessment of concrete production.

2.1.1. EF of Raw Material Use ($EF_m$)

The $EF_m$ is related to raw materials consumed for the concrete production. The $EF_m$ has been calculated by Equation (2) [5,21]:

$$\text{EFm} = \sum \left( \frac{(C_i + R_i) \cdot E_{mi}}{A_f / (1 - A_{oc})} \right) \cdot e_{CO2land} + \sum \left( \frac{C_{mi}}{Y_{mi}} \right) \cdot e_i \tag{2}$$

where $E_{mi}$ is the embodied emission of the material, and $C_i$ and $R_i$ are its material consumption and waste creation, respectively. One hectare of forest typically absorbs 2.7 t $CO_2$ on average [22], and $Aoc$ 0.30 t $CO_2$ is the fraction of annual oceanic emission sequestration [23], $C_{mi}$ is natural $i$th materials consumed in concrete production (tonnes) and $Y_{mi}$ represents the production yield factor of ith material (tonnes/ha). Table 1 lists the various types of bio-productive lands and the equivalency factor ($e_i$).

**Table 1.** Equivalence factor of different bio productive land [24].

| Bio Productive Land | Equivalence Factor $e_i$ (gha/ha) |
| --- | --- |
| $CO_2$ absorption land ($e_{CO_2\ land}$) | 1.28 |
| Forest land ($e_{forest\ land}$) | 1.28 |
| Crop land ($e_{cropland}$) | 2.52 |
| Pasture land ($e_{pasture\ land}$) | 0.43 |
| Sea productive/marine land ($e_{marine\ land}$) | 0.35 |

### 2.1.2. EF of Energy Consumption ($EF_e$)

The $EF_{me}$ is related to energy consumption for concrete production. The $EF_{me}$ of ready-mixed concrete is calculated by using Equation (3) [5]:

$$\text{EFme} = \sum (C_e \cdot \alpha_e) \cdot \left(\frac{1 - A_{oc}}{A_f}\right) \cdot e_{CO_{2land}} \tag{3}$$

where $C_e$ is the amount of direct energy used by machinery; $\alpha_e$ is the direct energy source's emission factor.

### 2.1.3. Labor ($EF_l$)

The ecological footprint of labor and human resources is related to metabolic rate [5,20]. The metabolic calories burned for different activity are mentioned in Table 2. The ecological footprint of labor/manpower for the concrete production is estimated as Equation (4) [5]:

$$\text{EF}_l = \left(\frac{EF_f}{365 \times daily\ calories\ burn}\right) \cdot \sum (M_r \cdot h_w)_i \tag{4}$$

where the $EF_f$ is the annual ecological footprint of food consumption. $M_r$ represents the metabolic rate of human activities (kcal/h). $h_w$ represents the working hours. The ecological footprint of food consumption for Indian people is about 0.549 gha/yr [5].

**Table 2.** The metabolic calories burned for different activity.

| Activity | Metabolic Rate (kcal/h) [25] | Avg. Ecological Footprint (gha/h) |
|---|---|---|
| High metabolic rate | 312.2–403.8 | $2.09 \times 10^{-4}$ |
| Moderate metabolic rate | 203.4–310.6 | $1.50 \times 10^{-4}$ |
| Low metabolic rate | 110.3–201.9 | $0.91 \times 10^{-4}$ |
| Machinery driver | 132.0–170.8 | $0.88 \times 10^{-4}$ |
| Crane operator | 100.9–225.2 | $0.95 \times 10^{-4}$ |
| Mason | 170.8–248.5 | $1.22 \times 10^{-4}$ |

The average daily consumption of calories per person in India is about 2576 kcal/capita/day [26,27].

### 2.1.4. Transportation ($EF_t$)

For the ready-mixed concrete plant, the transportation impact depends on raw materials transportation form their source to plant. The study considered cradle-to-gate approach for concrete production therefore transportation of final concrete to construction site has not been considered in this study. The estimation of $EF_T$ for the ready-mixed concrete is used as shown in Equation (5):

$$\text{EF}_t = \left(\sum \frac{X_{mi} \cdot T_{mi}}{C_{HDV}} \cdot T_{HDV}\right) \cdot \alpha_{fuel} \cdot \left(\frac{1 - A_{oc}}{A_f}\right) \cdot e_{CO_{2land}} \tag{5}$$

where $X_{mi}$ and $T_{mi}$ are material consumption and corresponding transportation distance of ith materials, respectively. $C_{HDV}$ (i.e., 3.5 tonnes) represents the capacity of $HDV$, respectively. $\alpha_{fuel}$ is the emission factor of fossil fuel (i.e., 3.17 $CO_2$ kg/kg diesel [28]), $T_{HDV}$ is the average fuel efficiency of $HDV$ (0.240 kg/km).

### 2.2. Ready-Mixed Concrete Materials

Concrete that has been pre-mixed with cement, sand, aggregates, and water is known as ready-mix concrete. At a centrally located batching plant, ready-mixed concrete is a type of concrete that is made in a factory to a predetermined recipe or to the customer's specifications. Concrete is brought to a job site, frequently in truck mixers equipped to

combine the concrete's ingredients, or immediately before the batch is delivered. This produces a precise mixture, enabling the development and use of specialty concrete mixtures on construction sites. In this study, the feasibility of partial replacement of cement with sun-dried and calcinated water treatment sludge (WTS) and sun-dried and calcinated stone dust has been investigated by using ecological footprint. The performance and economic assessment of substituting cement with stone dust and WTS in M20 grade (1:1.5:3 ratio as per the Indian standard) ready-mixed concrete have also been examined in this study. The IS sieve 90 μm was utilized to sieve the stone dust and WTS as per Indian specification standard for finding the fineness of cement, IS 460 (Part 1):1985 [29].

### 2.2.1. Stone Dust Properties

Kota stone is a sedimentary rock that can be found in a variety of colors and textures. Because the stone deposits are layered and scattered over well-formed weak planes, it is possible to create panels of stone with various thicknesses that are uniformly smooth and well-textured. Around 10–12 million tons of byproduct are produced annually by the flaggy limestone industry during the processing of stones. Around 10% of flaggy limestone powder and water make up the majority of the material. In order to promote the development of structures that are both sustainable and affordable, there is a tremendous demand for recycling the stone waste that is produced. The images of sun-dried and calcinated stone dust are shown in Figure 2a,b, respectively.

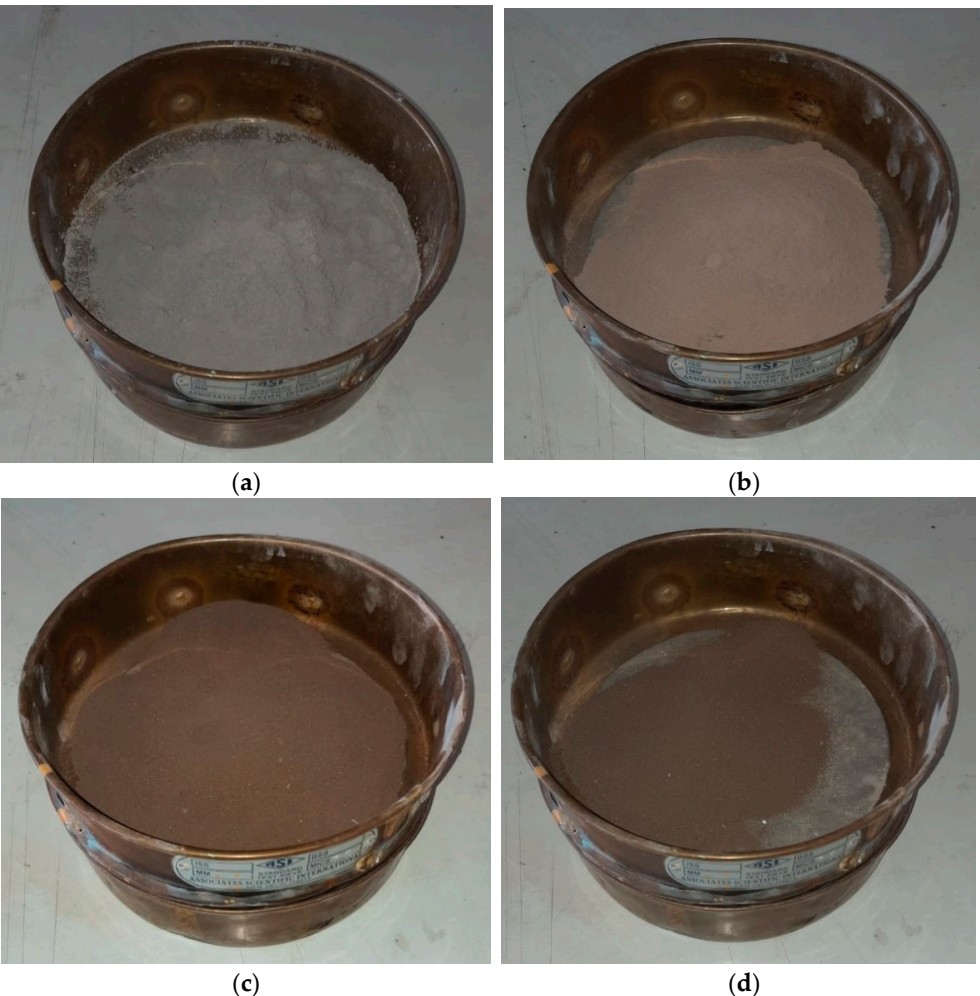

| | |
|:---:|:---:|
| (**a**) | (**b**) |
| (**c**) | (**d**) |

**Figure 2.** Images of (**a**) sun-dried stone dust, (**b**) calcinated stone dust, (**c**) sun-dried water treatment sludge, (**d**) calcinated water treatment sludge.

2.2.2. Water Treatment Sludge (WTS) Properties

WTS was sun-dried, therefore moisture content is found to be only 2.35%. The sludge has a low volatile matter concentration of 2.66% suggesting that it is inorganic in nature. The ash concentration of the sludge is 89.78%, whereas its loss on ignition is 8.96%. $SiO_2$ (52.78%), $Al_2O_3$ (14.38%), $Fe_2O_3$ (5.20%), and CaO (4.39%) are the main components of the sludge. The sludge listed in Table 3 also contains certain trace metals. In the dried sludge, several elements are present in alarmingly high concentrations. If WTS is not properly disposed of, barium, lead, arsenic, and other heavy metals could seriously harm the ecosystem. The images of sun-dried and calcinated water treatment sludge are shown in Figure 2c,d, respectively.

**Table 3.** The details of materials consumption in one-meter cubic concrete production in the plant.

| S. No | Materials | Unit | Quantity | | | | |
|---|---|---|---|---|---|---|---|
| | | | M20 | 5% Cement Replacement | 10% Cement Replacement | 15% Cement Replacement | 20% Cement Replacement |
| 1 | Cement | kg | 340 | 323 | 306 | 289 | 272 |
| 2 | Stone Dust | kg | - | 22.6 | 45.2 | 67.86 | 90.48 |
| 3 | Water treatment sludge | kg | - | 15.1 | 30.22 | 45.33 | 60.44 |
| 4 | Sand | kg | 941.16 | 941.16 | 941.16 | 941.16 | 941.16 |
| 5 | Aggregate 10 mm size | kg | 487.30 | 487.30 | 487.30 | 487.30 | 487.30 |
| 6 | Aggregate 20 mm size | kg | 493.61 | 493.61 | 493.61 | 493.61 | 493.61 |
| 7 | Water for mixing and workability | kg | 197 | 197 | 197 | 197 | 197 |
| 8 | Labor | days | 0.019 | 0.019 | 0.019 | 0.019 | 0.019 |
| 9 | Electricity consumption | kWh | 1.52 | 1.52 | 1.52 | 1.52 | 1.52 |

The details of the ready-mixed concrete (M20 grade, one-meter cubic concrete production) materials with partial replacement of cement with stone dust and water treatment sludge are mentioned in Table 3.

*2.3. Ecological Sustainability Index (ESI)*

A novel Ecological Sustainability Index (ESI) has been proposed in this study. It aims to address all the three pillars of sustainability: (i) environmental sustainability, (ii) social sustainability, and (iii) economic sustainability. Moreover, it may compare any form of goods as well as several industry kinds (such as small, medium, or large scale). ESI also prevents the issue of energy use allocation in multi-product manufacturing. The authors went on to quantitatively analyze the effects of ready-mixed concrete plants using the ESI concept.

The ESI is a simplified tool, which represents the socio-economic benefit of any type of industry per unit of its biocapacity consumption (i.e., ecological footprint). Biocapacity consumption for the industry or industrial products could be direct and indirect. While the fundamental nature of industrial production processes may be the cause of direct bio capacity consumption, human involvement in the production process and energy use is the cause of indirect bio capacity consumption. The purpose of the proposed ESI ratings is to evaluate the social, economic, and environmental objectives of any type or types of industries (i.e., small, medium, or large scale) The concept of ESI is illustrated through Figure 3. The system boundary for the estimation of ESI for an industry (or product) is shown in Figure 4.

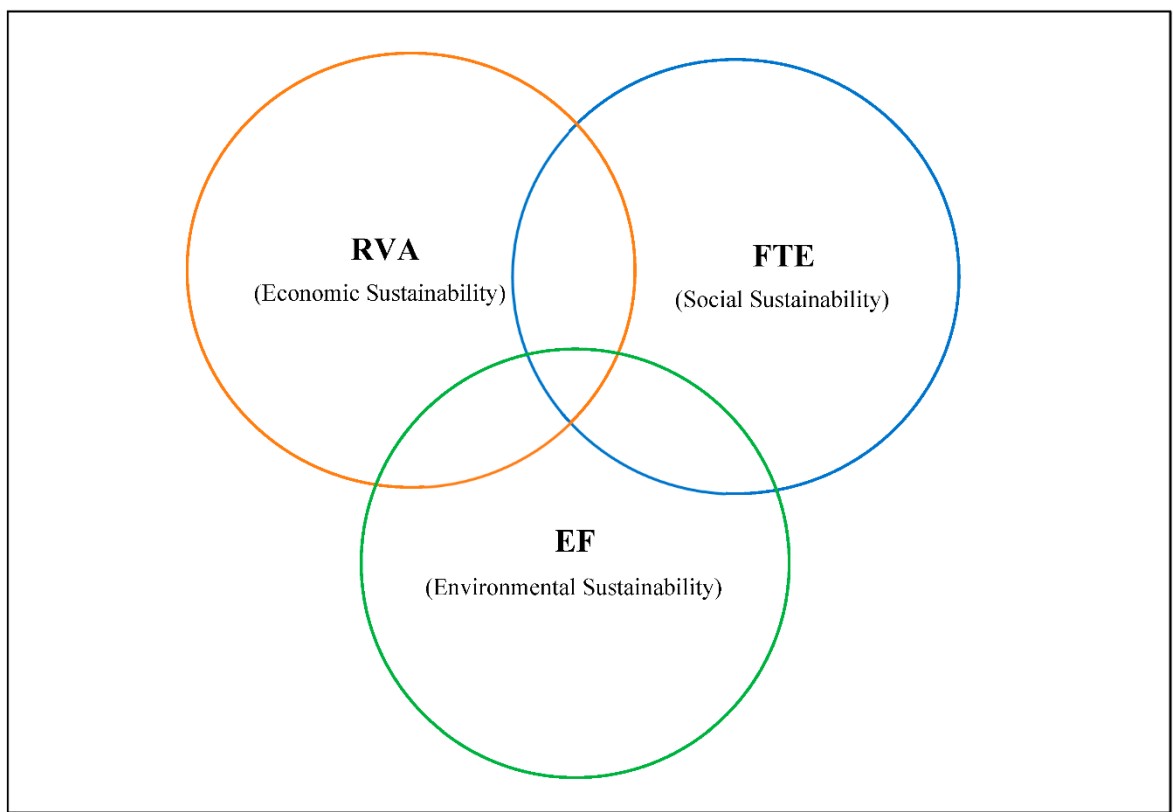

**Figure 3.** Conceptual representation of Ecological Sustainability Index for an industry (or product).

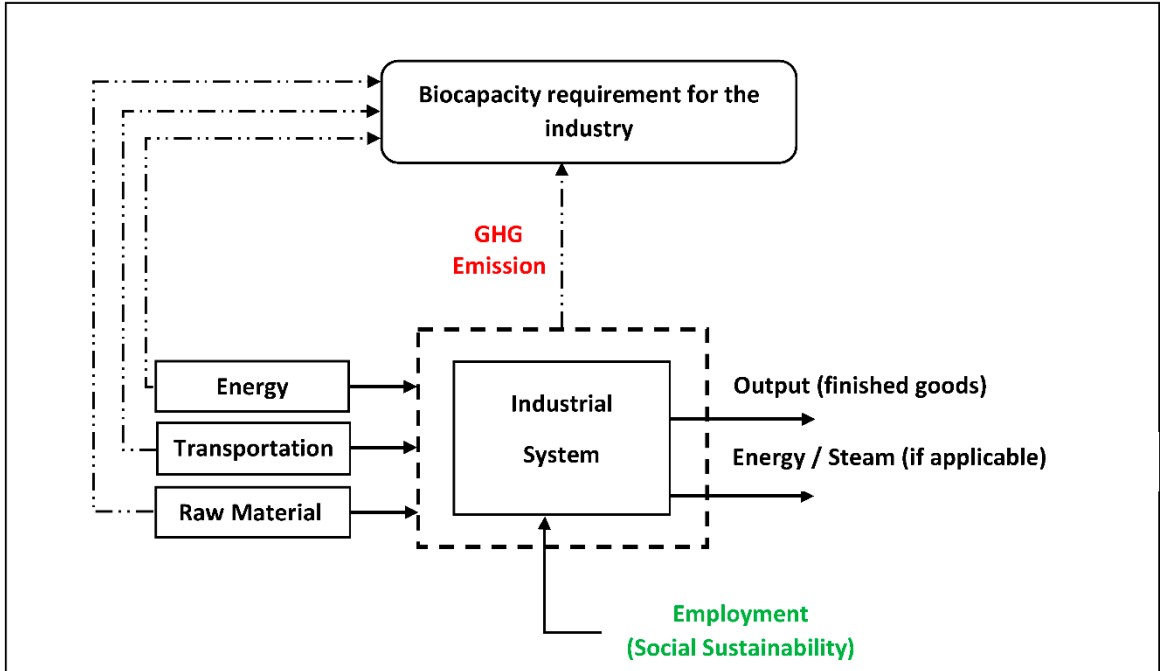

**Figure 4.** System boundary for the estimation of ESI for an industry (or product).

The expression of the ESI for an industry (or product) is as follows (Equation (6)):

$$\text{ESI} = \frac{RVA \cdot \left\{ FTE / (1 - r_{ue)} \right\}}{EF} \tag{6}$$

Here, the term "*RVA*" represents the resource value addition (i.e., the difference of the total annual economic values of outputs (i.e., products) of the industry and that of resource inputs (such as labor, energy cost, transportation, and land etc.,); it is represented here as $/year. The word "*FTE*" refers to all individuals employed by the sector during a given year. This relates to labor force that is employed full-time (i.e., 8 h per day for one year is equal to 1 *FTE*). It is possible to proportionally convert the manpower used for less than 8 h per day to the comparable *FTE*. For instance, 6 h a day of employment equivalent to 0.75 *FTE*. The phrase "$r_{ue}$" refers to the county's unemployment rate where the industry is located. The term *EF* represents the annual ecological footprint of an industry. It is an assessment tool that can be used to evaluate the viability of various sustainable solutions for the equitable distribution of the biocapacity of the region, nation, or planet. All resources, including energy, materials, water, and human activity, were taken into account during the analysis and converted into a single parameter (i.e., global hectare or gha).

## 3. Plant Description

In this case study, the ready-mixed plant is located at Malegaon city in India. The operational plant capacity is about 60 m³/h of concrete production. The detailed description of the ready-mixed concrete plant is mentioned in Table 4.

**Table 4.** The detailed description of the ready-mixed concrete plant.

| S. No. | Description | Details |
|---|---|---|
| 1 | Manufacturer | KYB Conmat |
| 2 | Capacity | 60 m$^3$/h |
| 3 | Nos. of in Line Bins | 4 |
| 4 | Charging Conveyor | Chevron Belt Conveyor |
| 5 | Cement storage capacity | 300 t |
| 6 | Water Weighing System | Yes |
| 7 | Admixture Weighing System | Available |
| 8 | Mixer | Twin Shaft |
| 9 | Maximum Size of Aggregate | 80 mm |
| 10 | Water Sprinkling & Gravity Discharge System | Yes |
| 11 | Air Compressor | Yes |
| 12 | Screw Conveyor for Cement | Yes |
| 13 | Control System | Automatic |
| 14 | Discharge Height | 4.1 m |
| 15 | Energy Consumption (for 60 m$^3$ concrete production) | 91 kWh (Grid electricity) or 12 L diesel/h (Diesel generator) |
| 16 | Manpower | 9 labor/day |
| 17 | Physical land | 8 hectares |
| 18 | Transportation of raw materials (from source to plant) | |
| | - Cement | 80–120 km |
| | - Sand | 10–20 km |
| | - Aggregate | 10–20 km |
| | - Stone Dust | 10–20 km |
| | - Water Treatment Sludge | 5–10 km |

## 4. Results and Discussions

The study focuses on environmental impact of ready-mixed conventional concrete as well as partial replacement of cement with waste materials such as stone dust and water treatment sludge in the concrete. Performance and economic assessment were also examined for ready-mixed concrete and compared with concrete using low impact materials (i.e., stone dust and water treatment sludge). The details of the assessment are as follows.

### 4.1. Ecological Footprint of Ready-Mixed Concrete ($EF_{RMC}$)

The EF of conventional ready-mixed concrete (M20 grade) production in the plant is estimated as 0.0295 gha/m$^3$. Sun-dried stone dust can replace 15% of cement by weight in the concrete with optimum (i.e., 22.82 N/mm$^2$) compressive strength. After replacing cement in the concrete with sun-dried stone dust up to 15% (by weight), the EF of concrete reduces up to 0.0256 gha/m$^3$ (i.e., 13.14% less than the conventional ready-mix concrete). Calcinated stone dust can replace 10% of cement by weight in the concrete with optimum (i.e., 24.65 N/mm$^2$) compressive strength. After replacing cement in the concrete with sun-dried stone dust up to 10% (by weight), the EF of concrete reduces up to 0.0294 gha/m$^3$ (i.e., 0.27% less than the conventional ready-mix concrete). The details of the EF of concrete with stone dust as the replacement materials are mentioned in Table 5.

Sun-dried water treatment sludge can replace 15% of cement by weight in the concrete with optimum (i.e., 22.56 N/mm$^2$) compressive strength. After replacing cement in the concrete with sun-dried water treatment sludge by up to 15% (by weight), the EF of concrete reduces up to 0.0255 gha/m$^3$ (i.e., 13.55% less than the conventional ready-mix concrete). Calcinated water treatment sludge can replace 10% of cement by weight in the concrete with optimum (i.e., 22.25 N/mm$^2$) compressive strength. After replacing cement in the concrete with sun-dried water treatment sludge up to 10% (by weight), the EF of concrete reduces up to 0.0286 gha/m$^3$ (i.e., 2.84% less than the conventional ready-mix concrete). The details of the EF of concrete with water treatment sludge as the replacement materials are mentioned in Table 6. The details of the EF parameters are discussed as follows:

Table 5. The details of the EF of concrete with sun-dried and calcinated stone dust as the replacement materials.

| S. No. | Description | Ecological Footprint (gha) | | | | | |
|---|---|---|---|---|---|---|---|
| | | Unit EF (gha) | M20 | 5% Cement Replacement | 10% Cement Replacement | 15% Cement Replacement | 20% Cement Replacement |
| 1 | Cement | $7.02 \times 10^{-5}$ [5] | 0.024 | 0.023 | 0.022 | 0.020 | 0.019 |
| 2 | Stone Dust | 0 | 0 | 0 | 0 | 0 | 0 |
| 3 | Sand | $1.57 \times 10^{-7}$ [5] | $1.48 \times 10^{-4}$ | $1.48 \times 10^{-4}$ | $1.48 \times 10^{-4}$ | $1.48 \times 10^{-4}$ | $1.48 \times 10^{-4}$ |
| 4 | Aggregate 10 mm size | $7.76 \times 10^{-8}$ [5] | $3.78 \times 10^{-5}$ | $3.78 \times 10^{-5}$ | $3.78 \times 10^{-5}$ | $3.78 \times 10^{-5}$ | $3.78 \times 10^{-5}$ |
| 5 | Aggregate 20 mm size | $7.76 \times 10^{-8}$ [5] | $3.83 \times 10^{-5}$ | $3.83 \times 10^{-5}$ | $3.83 \times 10^{-5}$ | $3.83 \times 10^{-5}$ | $3.83 \times 10^{-5}$ |
| 6 | Water for mixing and workability | $2.72 \times 10^{-7}$ [20] | $5.36 \times 10^{-5}$ | $5.36 \times 10^{-5}$ | $5.36 \times 10^{-5}$ | $5.36 \times 10^{-5}$ | $5.36 \times 10^{-5}$ |
| 7 | Labor | | $9.6 \times 10^{-6}$ | $9.6 \times 10^{-6}$ | $9.6 \times 10^{-6}$ | $9.6 \times 10^{-6}$ | $9.6 \times 10^{-6}$ |
| 8 | Electricity required to produce one $m^3$ concrete (i.e., 1.52 kWh/m$^3$) | $4.53 \times 10^{-6}$ | $1.48 \times 10^{-4}$ | $1.48 \times 10^{-4}$ | $1.48 \times 10^{-4}$ | $1.48 \times 10^{-4}$ | $1.48 \times 10^{-4}$ |
| 9 | Transportation | | | | | | |
| | - Cement | $7.21 \times 10^{-6}$ | $2.45 \times 10^{-3}$ | $2.23 \times 10^{-3}$ | $2.20 \times 10^{-3}$ | $2.08 \times 10^{-3}$ | $1.96 \times 10^{-3}$ |
| | - Sand | $1.08 \times 10^{-6}$ | $1.02 \times 10^{-3}$ | $1.02 \times 10^{-3}$ | $1.02 \times 10^{-3}$ | $1.02 \times 10^{-3}$ | $1.02 \times 10^{-3}$ |
| | - Aggregate (10 mm) | $1.08 \times 10^{-6}$ | $3.78 \times 10^{-5}$ | $3.78 \times 10^{-5}$ | $3.78 \times 10^{-5}$ | $3.78 \times 10^{-5}$ | $3.78 \times 10^{-5}$ |
| | - Aggregate (20 mm) | $1.08 \times 10^{-6}$ | $3.83 \times 10^{-5}$ | $3.83 \times 10^{-5}$ | $3.83 \times 10^{-5}$ | $3.83 \times 10^{-5}$ | $3.83 \times 10^{-5}$ |
| | - Stone Dust | $1.08 \times 10^{-6}$ | 0.000 | $2.33 \times 10^{-5}$ | $4.65 \times 10^{-5}$ | $6.97 \times 10^{-5}$ | $9.31 \times 10^{-5}$ |
| **Total EF of ready-mixed concrete using sun-dried stone dust** | | | **0.0295** | **0.0282** | **0.0269** | **0.0256** | **0.0243** |
| **EF for Calcination of Stone Dust** | | | 0 | $1.27 \times 10^{-3}$ | $2.5 \times 10^{-3}$ | $3.75 \times 10^{-3}$ | $5.53 \times 10^{-3}$ |
| **Total EF of concrete** | | | **0.0295** | **0.0295** | **0.0294** | **0.0293** | **0.0298** |

EF of 1 MJ is $1.94 \times 10^{-5}$ gha [21], Energy Required for Calcination is about 3 GJ/ton, Emission factor 90.6 gCO$_2$/MJ for Coal [30].

**Table 6.** The details of the EF of concrete with sun-dried and calcinated water treatment sludge as the replacement materials.

| S. No. | Description | | Ecological Footprint (gha) | | | | |
|---|---|---|---|---|---|---|---|
| | | Unit EF (gha) | M20 | 5% Cement Replacement | 10% Cement Replacement | 15% Cement Replacement | 20% Cement Replacement |
| 1 | Cement | $7.02 \times 10^{-5}$ | 0.024 | 0.023 | 0.022 | 0.020 | 0.019 |
| 2 | Stone Dust | 0 | 0 | 0 | 0 | 0 | 0 |
| 3 | Sand | $1.57 \times 10^{-7}$ | $1.48 \times 10^{-4}$ | $1.48 \times 10^{-4}$ | $1.48 \times 10^{-4}$ | $1.48 \times 10^{-4}$ | $1.48 \times 10^{-4}$ |
| 4 | Aggregate 10 mm size | $7.76 \times 10^{-8}$ | $3.78 \times 10^{-5}$ | $3.78 \times 10^{-5}$ | $3.78 \times 10^{-5}$ | $3.78 \times 10^{-5}$ | $3.78 \times 10^{-5}$ |
| 5 | Aggregate 20 mm size | $7.76 \times 10^{-8}$ | $3.83 \times 10^{-5}$ | $3.83 \times 10^{-5}$ | $3.83 \times 10^{-5}$ | $3.83 \times 10^{-5}$ | $3.83 \times 10^{-5}$ |
| 6 | Water for mixing and workability | $2.72 \times 10^{-7}$ | $5.36 \times 10^{-5}$ | $5.36 \times 10^{-5}$ | $5.36 \times 10^{-5}$ | $5.36 \times 10^{-5}$ | $5.36 \times 10^{-5}$ |
| 7 | Labor | - | $9.6 \times 10^{-6}$ | $9.6 \times 10^{-6}$ | $9.6 \times 10^{-6}$ | $9.6 \times 10^{-6}$ | $9.6 \times 10^{-6}$ |
| 8 | Electricity required to produce one m$^3$ concrete (i.e., 1.52 kWh/m$^3$) | $4.53 \times 10^{-6}$ | $1.48 \times 10^{-4}$ | $1.48 \times 10^{-4}$ | $1.48 \times 10^{-4}$ | $1.48 \times 10^{-4}$ | $1.48 \times 10^{-4}$ |
| | Transportation | | | | | | |
| 9 | - Cement | $7.21 \times 10^{-6}$ | $2.45 \times 10^{-3}$ | $2.23 \times 10^{-3}$ | $2.20 \times 10^{-3}$ | $2.08 \times 10^{-3}$ | $1.96 \times 10^{-3}$ |
| | - Sand | $1.08 \times 10^{-6}$ | $1.02 \times 10^{-3}$ | $1.02 \times 10^{-3}$ | $1.02 \times 10^{-3}$ | $1.02 \times 10^{-3}$ | $1.02 \times 10^{-3}$ |
| | - Aggregate (10 mm) | $1.08 \times 10^{-6}$ | $3.78 \times 10^{-5}$ | $3.78 \times 10^{-5}$ | $3.78 \times 10^{-5}$ | $3.78 \times 10^{-5}$ | $3.78 \times 10^{-5}$ |
| | - Aggregate (20 mm) | $1.08 \times 10^{-6}$ | $3.83 \times 10^{-5}$ | $3.83 \times 10^{-5}$ | $3.83 \times 10^{-5}$ | $3.83 \times 10^{-5}$ | $3.83 \times 10^{-5}$ |
| | - WTS | $1.08 \times 10^{-6}$ | 0 | $1.63 \times 10^{-5}$ | $3.27 \times 10^{-5}$ | $4.90 \times 10^{-5}$ | $6.54 \times 10^{-5}$ |
| | **Total EF of Concrete using sun-dried WTS** | | **0.0295** | **0.0282** | **0.0269** | **0.0255** | **0.0243** |
| | **EF for Calcination of WTS** | | 0 | $8.79 \times 10^{-5}$ | $1.76 \times 10^{-3}$ | $2.64 \times 10^{-3}$ | $3.52 \times 10^{-3}$ |
| | **TOTAL EF of concrete using calcinated WTS** | | **0.0295** | **0.0290** | **0.0286** | **0.0282** | **0.0278** |

### 4.1.1. EF of Raw Material Use (EF$_m$)

The EF of raw materials use in ready-mixed conventional M20 grade concrete is estimated as 0.0246 gha/m$^3$ (i.e., 83.2% of the total EF of the conventional ready-mixed concrete). The material impact is the highest among all the four EF parameters of concrete. For stone dust as a replacement of cement: (i) sun-dried stone dust concrete has 0.021 gha/m$^3$ (i.e., 81.8% environmental impact of the total environmental impact of the concrete replacing 15% by weight of cement); (ii) calcinated stone dust concrete has 0.0222 gha/m$^3$ (i.e., 75.3% environmental impact of the total environmental impact of the concrete replacing 10% by weight of cement). For water treatment sludge as a replacement of cement: (i) sun-dried water treatment sludge concrete has 0.021 gha/m$^3$ (i.e., 82.2% environmental impact of the total environmental impact of the concrete replacing 15% by weight of cement); (ii) calcinated water treatment sludge concrete has 0.0222 gha/m$^3$ (i.e., 77.3% environmental impact of the total environmental impact of the concrete replacing 10% by weight of cement).

### 4.1.2. EF of Energy Consumption (EF$_e$)

The EF of energy consumption for M20 grade ready-mixed conventional concrete production is estimated as 0.0004 gha/m$^3$ (i.e., 1.4% of the total impact of the conventional concrete). The EF of energy consumption of ready-mixed concrete using calcinated stone dust (10% by weight of cement replacement with calcinated stone dust) is estimated as 9.9% of the total EF of the concrete. However, only 1.6% environmental impact of the total impact of the concrete (15% by weight of cement replacement with sun-dried stone dust) has been examined for the use of sun-dried stone dust. It is due to less energy involvement in sun-dried process of the stone dust as compared to calcination process of stone dust. It indicates that the use of sundried stone dust in the concrete is more effective than calcinated stone dust.

The EF of energy consumption of ready-mixed concrete using calcinated water treatment sludge (10% by weight of cement replacement with calcinated water treatment sludge) is estimated as 7.6% of the total EF of the concrete. However, only 1.6% environmental impact of the total impact of the concrete (15% by weight of cement replacement with sun-dried water treatment sludge) has been examined for the use of sun-dried water treatment sludge. It is due to less energy involvement in sun-dried process of the water treatment sludge as compared to calcination process of water treatment sludge. It indicates that the use of sun-dried water treatment sludge in the concrete is more effective than calcinated water treatment sludge.

### 4.1.3. Labor (EF$_l$)

The EF for labor is similar for all the proposed types of ready-mixed concrete (i.e., stone dust and water treatment sludge for calcination and sun-dried) as well as conventional concrete production in the plant. The study neglects the labor impact due to sun-drying and calcination processes of stone dust and water treatment sludge. The estimated EF of labor is about $9.6 \times 10^{-6}$ gha per m$^3$ ready-mixed concrete production.

### 4.1.4. Transportation (EF$_t$)

The EF of transportation of materials for M20 grade ready-mixed conventional concrete production is estimated as 0.0045 gha/m$^3$ (i.e., 15.4% of the total impact of the conventional concrete). The EF of transportation of ready-mixed concrete using calcinated stone dust (10% by weight of cement replacement with calcinated stone dust) is estimated as 14.7% of the total EF of the concrete. However, 16.5% environmental impact of the total impact of the concrete (15% by weight of cement replacement with sun-dried stone dust) was examined for the use of sun-dried stone dust. It is due to the less energy involvement in sun-dried process of the stone dust as compared to the calcination process of stone dust. It indicates that the use of sun-dried stone dust in the concrete is more effective than calcinated stone dust. The EF of transportation of materials for sun-dried water treatment sludge at 15% by weight of cement replacement is the same as the sun-dried stone dust that is 16.5%

environmental impact of the total impact of the concrete, whereas for the calcinated water treatment sludge at 10% (by weight) is found 15.1%.

### 4.2. Performance and Economic Assessment

The experimental research work examined the compressive strength of M20 ready-mixed concrete and partial replacement of cement with stone dust (sun-dried and calcinated) and water treatment sludge (sun-dried and calcinated) in the concrete. Figure 5 shows the failure modes of concrete cube samples with (a) partial cement replacement by stone dust, (b) partial cement replacement by WTS, (c) conventional concrete, and (d) compression testing machine (CTM). It is found that up to 15% (by weight) cement replacement with sun-dried stone dust can give compressive strength (i.e., 22.82 N/mm$^2$) more than the targeted compressive strength (i.e., 20 N/mm$^2$) of conventional M20 grade concrete. However, calcinated stone dust gives compressive strength (i.e., 20.7 N/mm$^2$) very close to the targeted compressive strength. The compressive strength of different concrete cubes is shown in Figure 6. The experimental results suggested that the calcinated stone dust can feasibly replace cement up to 10% (by weight) in the conventional M20 grade concrete. Similarly, for the use of water treatment sludge as cement replacement material in the M20 concrete, the sun-dried water treatment sludge can successfully replace cement up to 15% by weight in the concrete with satisfactory compressive strength (i.e., 22.56 N/mm$^2$). However, the calcinated water treatment sludge can replace cement up to 10% by weight in the concrete. Various studies reported similar results that the contents of calcined sludge in cementitious matrix up to 15 wt% is possible. However, for the best mechanical results the feasible sludge addition in cementitious matrix is up to 10% by weight [31–33]. Panesar and Zhang [27] concluded that the replacement levels of 5–15% are effective in observing decreased chloride permeability when compared to the reference concrete.

The cost of M20 grade ready-mixed concrete is nearly USD 72.78 per m$^3$ in India (USD 1 = INR 82.78). The cost of sun-dried stone dust and water treatment sludge for 15% (by weight) cement replacement in the concrete is estimated as 68.60 \$/m$^3$ and 68.40 \$/m$^3$, respectively. The cost of calcinated stone dust and water treatment sludge for 10% cement replacement in the concrete is estimated as 70.60 \$/m$^3$ and 70.24 \$/m$^3$, respectively. The ready-mixed M20 grade concrete production cost can be reduced up to 6% by using sundried water treatment sludge in the conventional concrete. The detailed economic assessment of sun-dried and calcinated stone dust used in ready-mixed concrete is mentioned in Table 7. The detailed economic assessment of sun-dried and calcinated water treatment sludge used in ready-mixed concrete is mentioned in Table 8.

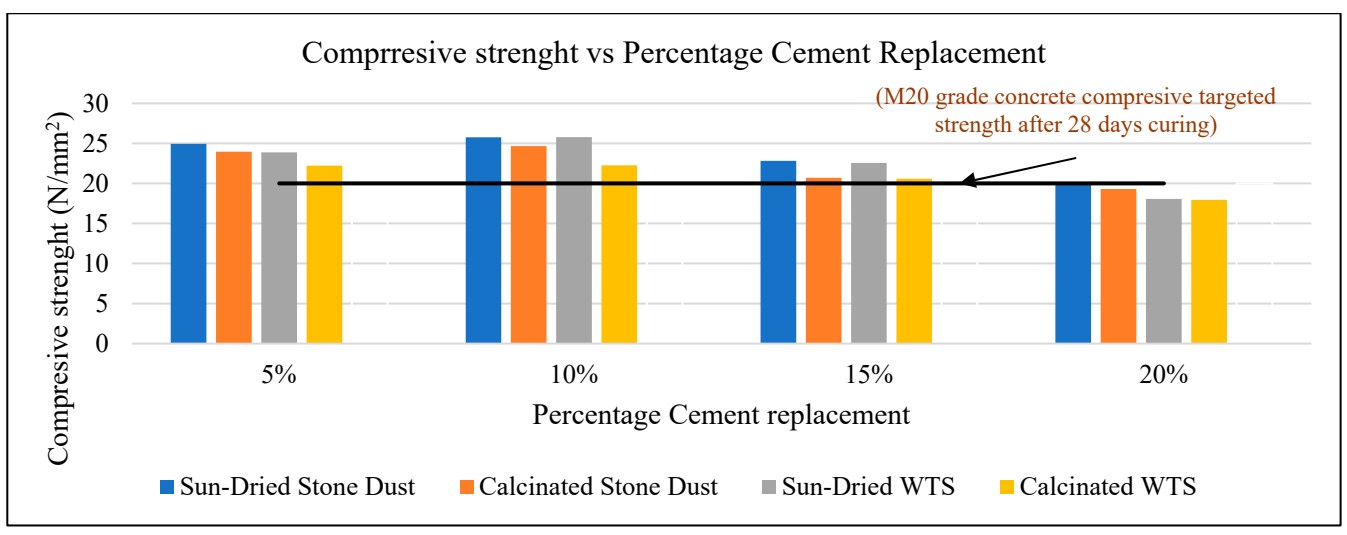

**Figure 5.** Failure modes of concrete cube samples with (**a**) partial cement replacement by stone dust, (**b**) partial cement replacement by WTS, (**c**) conventional concrete, and (**d**) compression testing machine (CTM).

**Figure 6.** Compressive strength of concrete cube for different proportion of cement replacement with stone dust and water treatment sludge.

**Table 7.** The detailed economic assessment of sun-dried and calcinated stone dust used in ready-mixed concrete.

| S. No. | Description | Cost (USD) | | | | | |
|:---:|:---|:---:|:---:|:---:|:---:|:---:|:---:|
| | | Unit Cost (USD) | M20 | 5% Cement Replacement | 10% Cement Replacement | 15% Cement Replacement | 20% Cement Replacement |
| 1 | Cement | 0.085/kg | 28.9 | 27.45 | 26.01 | 24.56 | 23.12 |
| 2 | Stone Dust | 3.02/ton | 0 | 0.068 | 0.136 | 0.205 | 0.273 |
| 3 | Sand | 0.0088/kg | 8.28 | 8.28 | 8.28 | 8.28 | 8.28 |
| 4 | Aggregate 10 mm size | 0.014/kg | 6.82 | 6.82 | 6.82 | 6.82 | 6.82 |
| 5 | Aggregate 20 mm size | 0.014/kg | 6.91 | 6.91 | 6.91 | 6.91 | 6.91 |
| 6 | Water for mixing and workability | $0.088/m^3$ | 17.33 | 17.33 | 17.33 | 17.33 | 17.33 |
| 7 | Labor | 6.28/day | 0.12 | 0.12 | 0.12 | 0.12 | 0.12 |
| 8 | Electricity required to produce one $m^3$ concrete (i.e., 1.52 $kWh/m^3$) | 0.102/kWh | 0.156 | 0.156 | 0.156 | 0.156 | 0.156 |
| 9 | Transportation | 0.027 $/ton-km | | | | | |
| | - Cement | | 0.92 | 0.87 | 0.82 | 0.78 | 0.73 |
| | - Sand | | 2.03 | 2.03 | 2.03 | 2.03 | 2.03 |
| | - Aggregate (10 mm) | | 0.66 | 0.66 | 0.66 | 0.66 | 0.66 |
| | - Aggregate (20 mm) | | 0.66 | 0.66 | 0.66 | 0.66 | 0.66 |
| | - Stone Dust | | 0.00 | 0.03 | 0.06 | 0.09 | 0.12 |
| **Total cost of ready-mixed concrete using sun-dried stone dust** | | | **72.78** | **71.316** | **69.856** | **68.60** | **66.936** |
| **Cost for Calcination of Stone Dust** | | | **0** | **0.29** | **0.58** | **0.87** | **1.16** |
| **Total cost of concrete using calcinated stone dust** | | | **72.78** | **71.67** | **70.57** | **69.5** | **68.36** |

EF of 1 MJ is $1.94 \times 10^{-5}$ gha [21], Energy Required for Calcination is about 3 GJ/ton, Emission factor 90.6 $gCO_2/MJ$ for Coal [30] Coal cost 108.6 $ / ton (i.e., 24 MJ/kg).

**Table 8.** The detailed economic assessment of sun-dried and calcinated water treatment used in ready-mixed concrete.

| S. No. | Description | Unit Cost (USD) | Cost (USD) | | | | |
|---|---|---|---|---|---|---|---|
| | | | M20 | 5% Cement Replacement | 10% Cement Replacement | 15% Cement Replacement | 20% Cement Replacement |
| 1 | Cement | 0.085/kg | 28.9 | 27.45 | 26.01 | 24.56 | 23.12 |
| 2 | WTS | 0 | 0 | 0.068 | 0.136 | 0.205 | 0.273 |
| 3 | Sand (kg) | 0.0088/kg | 8.28 | 8.28 | 8.28 | 8.28 | 8.28 |
| 4 | Aggregate 10 mm size | 0.014/kg | 6.82 | 6.82 | 6.82 | 6.82 | 6.82 |
| 5 | Aggregate 20 mm size | 0.014/kg | 6.91 | 6.91 | 6.91 | 6.91 | 6.91 |
| 6 | Water for mixing and workability | 0.088/m$^3$ | 17.33 | 17.33 | 17.33 | 17.33 | 17.33 |
| 7 | Labor | 6.28/day | 0.12 | 0.12 | 0.12 | 0.12 | 0.12 |
| 8 | Electricity required to produce one m$^3$ concrete (i.e., 1.52 kWh/m$^3$) | 0.102/kWh | 0.156 | 0.156 | 0.156 | 0.156 | 0.156 |
| 9 | Transportation | 0.027 $/ton-km | | | | | |
| | - Cement | | 0.92 | 0.87 | 0.82 | 0.78 | 0.73 |
| | - Sand | | 2.03 | 2.03 | 2.03 | 2.03 | 2.03 |
| | - Aggregate (10 mm) | | 0.66 | 0.66 | 0.66 | 0.66 | 0.66 |
| | - Aggregate (20 mm) | | 0.66 | 0.66 | 0.66 | 0.66 | 0.66 |
| | - WTS | | 0.00 | 0.03 | 0.06 | 0.09 | 0.12 |
| **Total cost of ready-mixed concrete using sun-dried WTS** | | | **72.786** | **71.316** | **69.856** | **68.396** | **66.936** |
| **Cost for Calcination of WTS** | | | 0 | 0.203 | 0.407 | 0.61 | 0.812 |
| **Total cost of concrete using calcinated WTS** | | | **72.78** | **71.50** | **70.24** | **68.97** | **67.70** |

*4.3. Ecological Sustainability Index (ESI) of the Ready-Mixed Plant*

The cost (resource inputs) of conventional ready-mixed concrete is about 72.78 $/m$^3$ and the plant sells the concrete at 75 $/m$^3$. It means RVA of the plant is 2.22 $/m$^3$ and net annual revenue of the plant is USD 265,680. The FTE of the plant is 9 and $r_{ue}$ (unemployment rate of India) is 0.06 [34]. The annual ecological footprint of the case study plant is 3540.73 gha (for 120,000 m$^3$ of concrete production per year). The ESI of the ready-mixed concrete plant is estimated as 718.42 $/gha. It indicates that for each bio-productive land consumption, the socio-economic benefit of the case study ready-mixed concrete plant is about USD 718.42. The ESI of the ready-mixed concrete plant may improve by using stone dust and water treatment sludge in the concrete.

**5. Conclusions**

In this study, experimental investigations were carried out on conventional RMC (M20 grade) and RMC (M20 grade) with partial cement replacement (i.e., 5%, 10%, 15%, and 20% by weight) by stone dust and water treatment sludge. The results showed that the sun-dried water treatment sludge and sun-dried stone dust could effectively replace cement (15% by weight) in the concrete, with a comparable compressive strength over the M20 ready-mixed concrete. However, the experimental results (based on mechanical results) suggested that the calcinated stone dust and WTS can feasibly replace cement up to 10% (by weight) in the conventional M20 grade concrete. A novel sustainability index has also been proposed for the RMC plant to classify the different types of concrete based on environmental, social, and economic sustainability.

The comparison of conventional RMC with stone dust and water treatment sludge leads to the following conclusions:

- The EF of conventional RMC was estimated as 0.0295 gha/m$^3$. Its compressive strength was 23.93 N/mm$^2$, and the production cost was 72.78 $/m$^3$.
- The EF of RMC for 15% cement replacement with sun-dried stone dust was 0.0256 gha/m$^3$ (i.e., 13.14% less than the conventional ready-mix concrete). Its compressive strength was 22.82 N/mm$^2$, and the production cost was 68.60 $/m$^3$.
- The EF of RMC for 10% cement replacement with calcinated stone dust was 0.0294 gha/m$^3$ (i.e., 0.27% less than the conventional RMC). Its compressive strength was 24.65 N/mm$^2$, and the production cost was 70.57 $/m$^3$.
- The EF of RMC for 15% cement replacement with sun-dried WTS was 0.0255 gha/m$^3$ (i.e., 8.81% less than the total EF of conventional RMC). Its compressive strength was 30.78 N/mm$^2$, and the production cost was 69.4 $/m$^3$.
- The EF of RMC for the 10% cement replacement with calcinated WTS was 0.0287 gha/m$^3$ (i.e., 4.26% less than the total EF of conventional RMC). Its compressive strength was 22.58 N/mm$^2$, and the production cost was 70.2 $/m$^3$.

The performance of sun-dried stone dust and WTS presents better results; however, calcinated stone dust and WTS are recommended for concrete production to avoid the unwanted presence of moisture, which can reduce the material's shelf life and alter the desired workability. The use of waste (stone dust and water treatment sludge) in the concrete may also improve the ESI of the ready-mixed plant. The study suggested that the ecological footprint of concrete reduces by using (sun-dried and calcinated) stone dust and WTS as cement replacement. Using low-environmental and low-cost materials may also improve the ESI of a construction industry.

**Author Contributions:** Conceptualization, D.H., Y.A. and R.P.; methodology, D.H.; validation, U.K.D., J.H. and M.H.; formal analysis, Y.A. and N.A.K.; investigation, N.A.K.; resources, Y.A.; data curation, N.A.K.; writing—original draft preparation, Y.A. and N.A.K.; writing—review and editing, N.A.K. and Y.A.; supervision, D.H. and R.P.; project administration, D.H. All authors have read and agreed to the published version of the manuscript.

**Funding:** This research received no external funding.

**Institutional Review Board Statement:** Not applicable.

**Informed Consent Statement:** Informed consent was obtained from all subjects involved in the study.

**Data Availability Statement:** Suggested Data Availability Statements are available in section "MDPI Research Data Policies" at https://www.mdpi.com/ethics.

**Conflicts of Interest:** The authors declare no conflict of interest.

## Abbreviations

| | |
|---|---|
| $C_{HDV}$ | Capacity of heavy-duty vehicle |
| $\alpha_e$ | Direct energy source's emission factor |
| $C_e$ | Direct energy used by machinery |
| $A_f$ | Absorption factor |
| $A_{oc}$ | Annual oceanic emission |
| $C_i$ | Material consumption |
| $C_{mi}$ | Natural materials consumed in concrete |
| $EF_e$ | Ecological Footprint of energy consumption in the concrete plant. |
| $EF_f$ | Annual Ecological Footprint of food consumption |
| $EF_l$ | Ecological Footprint of labor in concrete production |
| $EF_l$ | Ecological Footprint of labor |
| $EF_m$ | Ecological Footprint of raw materials use in concrete |
| $EF_{me}$ | Energy consumption |
| $EF_{RMC}$ | Ecological footprint of ready-mixed concrete |
| $EF_t$ | Ecological Footprint for transportation of raw materials from their source |
| $E_i$ | Bio-productive lands and the equivalency factor |
| $E_{mi}$ | Embodied emission of the material |
| ESI | Ecological sustainability index |
| gha | Global hector |
| GHG | Greenhouse gas |
| LCA | Life cycle assessment |
| $M_r$ | Metabolic rate of human activities |
| OPC | Ordinary Portland cement |
| PM | Particulate matter |
| RAC | Recycled aggregate concrete |
| $R_i$ | Waste creation |
| $r_{ue}$ | County's unemployment rate |
| $T_{HDV}$ | Average fuel efficiency of HDV |
| $T_{mi}$ | Transportation distance |
| WTS | Water treatment sludge |
| $X_{mi}$ | Materials consumption |
| $Y_{mi}$ | Production yield factor |
| $\alpha_{fuel}$ | Emission factor of fossil fuel |

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
