# Peer review of "Ecological Footprint Assessment of Concrete: Partial Replacement of Cement by Water Treatment Sludge and Stone Dust"

_sustainability, doi:10.3390/su15097512_

Round 1
Reviewer 1 Report
In this paper, Water Treatment Sludge & Stone Dust are used to replace part of the cement, the topic is fascinating, and the scope is suitable for the journal. However, there are some problems in this paper's writing and experimental analysis, and it is recommended after major revision to reconsider.
Abstract: The author only focuses on changing mechanical properties; is the sludge harmful? Is there any dissolution of heavy metal ions?
The relevant conclusions are not reflected in the abstract of whether it will cause secondary pollution to the environment.
Line52-54, For a background discussion of CO2 emissions, citing the latest literature,these two kinds of literature can be referred to
(https://doi.org/10.1016/j.jobe.2022.104880,https://doi.org/10.1016/j.jobe.2023.106018)
Line69-70,It is necessary to introduce the common methods of developing low-carbon building materials in detail, including the use of low-carbon cement, such as sulfoaluminate cement, to replace part of the cement and the utilization of solid waste, including the development of low -calcium cement to improve durability and indirectly reduce carbon emissions.
Please refer to this literature. https://doi.org/10.1016/j.conbuildmat.2022.126921
Line94-103: Is it necessary to consider the study of microscopic properties after replacing part of cement?
For materials , Please introduce the chemical composition and physical properties of Water Treatment Sludge & Stone Dust in detail.
For Fig.4,The FIG is too random, replace or delete.
Line247-250, Now that the environmental impact has been studied, why is there no research on the possibility of secondary pollution from Water Treatment Sludge & Stone Dust?
The figures and tables in the full text must be redrawn; it is too random to be a scientific paper.
Author Response
Dear Reviewer, Thank you for your suggestions. We have tried to incorporate all your possible suggestions in the revised manuscript.

Reviewer 2 Report
Flaws with regard to the English language have to be addressed by performing thorough proofreading of the whole draft manuscript and consulting a professional English proofreader.
More explanation on the gap in present research and novelty of the study in connection to the motivation to develop a novel Ecological Sustainability Index has to be included in the Abstract.
The explanation that introduces water treatment sludge and the motivation for its adoption in concrete has to be enhanced.
Information presented from line 71 to 75 has to be supported with citations from relevant and recent sources, preferably those that were published since 2018.
References [8], [9], [10], [11], [13], [15], [16], [17] and [18] have to be replaced with those that are more recent, preferably those that were published since 2018. Information that is presented with the support of those references has to be updated accordingly.
Discussion of results has to be performed with reference to the sources cited throughout the literature review as presented in the Introduction section.
Recommendations for future research have to be presented after stating the conclusions.
Author Response

(The authors gave the same response as above.)

Reviewer 3 Report
The authors tried to decrease use of the cement with waste materials. The paper is generally good but it needs improvement. Followings should be carried out before acceptance:
The abstract should contain important results of the study.
How this recycled materials for this study is obtained?
Add sieve analysis results in Figure.
What are the chemical properties of cement?
Novelty is not clear. Very same studies are already exists. What is the difference?
The reason for selecting design mixture should be added.
Compare your results with existing studies
Other types of powder can also be used as cementitious materials such as glass powder and coal bottom ash. For this purpose the following studies also should be add:
[1] influence of replacing cement with waste glass on mechanical properties of concrete
[2] use of recycled coal bottom ash in reinforced concrete beams as replacement for aggregate
[3] concrete containing waste glass as an environmentally friendly aggregate: a review on fresh and mechanical characteristics;
[4] mechanical behavior of crushed waste glass as replacement of aggregates;
[5] flexural behavior of reinforced concrete beams using waste marble powder towards application of sustainable concrete
Add photos for test setup?
Add photos for utilized materials. There is no photo related to which materials are utilized.
Please add damaged photos damaged photos of samples
Add some summary for conclusion
Add recent studies on this subject to introduction. There are many studies on the introduction for this topic.
Conclusion should be improved. The recommendation considering all test should be given for engineers.
Author Response

(The authors gave the same response as above.)

Reviewer 4 Report
Thank you for your effort and your work need to improve like:
1- using sludge with sun dried is not effective, it must be burn up to 100 degree at least.
2- it must be make more microstructure tests like, SEM, EDX, etc.
Author Response

(The authors gave the same response as above.)

Reviewer 5 Report
This article discusses the ecological footprint assessment of concrete and the partial substitution of cement with sludge from water treatment. In this research, the authors attempt to evaluate the environmental impact of the manufacturing of concrete with supplemental cementitious material for standard concrete by employing the ecological footprint and a novel sustainability index for ready-mixed concrete plants. The article is well-written and can be published with a few minor edits. Please see the comments listed below.
The abstract of the article includes some of the article's introduction. Please erase any redundant or non-abstract content, such as the opening few phrases of the abstract, and condense your abstract for improved readability.
Ensure that your article includes a short introduction that establishes the topic and what the reader can expect from the remainder of the content.
Make sure to define any technical terms you use so that your readers may understand them.
Useful would be background information on the ecological footprint of concrete and why it is so important to reduce that footprint. Please elaborate on the benefits of employing sewage sludge and stone dust as partial cement replacements in concrete.
5.Conclude by discussing the implications of your findings to the construction industry and the natural environment. Similarly, make sure to highlight any limitations or unknowns in your study. Restate the significance of your findings and underline the need to reduce the negative environmental impact of concrete.
Following the above comments, this article can be published.
Author Response

(The authors gave the same response as above.)

Reviewer 6 Report
It is required to mention the references of equations 1, 3, 4, 5
Do the quantities of stone dust and water treatment sludge in Table 1 account for the volume or weight replacement of cement?
What is your basis for the Ecological Sustainability Index being novel?
Check the unit of water for mixing and workability in the unit cost of Tables 7 and 8
It is required to mention the meaning of all abbreviations used in the equations such as Aoc, Af
Author Response

(The authors gave the same response as above.)

Round 2
Reviewer 1 Report
The author has revised the manuscript according to the reviewer's comments, and the quality of the manuscript has been improved. It is recommended to accept it after minor revision.
Line55-56,For CO2 data, it is recommended to add the latest references,such as,10.1016/j.cemconcomp.2023.105071.
Author Response
Dear Reviewer,
Thank you for your constructive suggestion. Your suggestion has been incorporated into the revised manuscript.
With regards,
Reviewer 3 Report
The paper can be accepted
Author Response
Dear Reviewer,
Thank you for accepting our manuscript for publication.
With regards
Reviewer 4 Report
Thank you for your response
Author Response
Dear Reviewer,
Thank you for accepting our manuscript for publication.
With regards,